# Reusable Nano-Zirconia-Catalyzed Synthesis of Benzimidazoles and Their Antibacterial and Antifungal Activities

**DOI:** 10.3390/molecules26144219

**Published:** 2021-07-12

**Authors:** Tentu Nageswara Rao, Suliman Yousef AlOmar, Faheem Ahmed, Fadwa Albalawi, Naushad Ahmad, Nalla Krishna Rao, M. V. Basaveswara Rao, Ravi Kumar Cheedarala, G. Rajasekhar Reddy, Tentu Manohra Naidu

**Affiliations:** 1Department of Chemistry, Krishna University, Machilipatnam 521001, Andhra Pradesh, India; tnraochemistry@gmail.com; 2Department of Zoology, College of Science, King Saud University, Riyadh 11451, Saudi Arabia; fadwa_saad@hotmail.com; 3Department of Physics, College of Science, King Faisal University, Hofuf 31982, Al-Ahsa, Saudi Arabia; fahmed@kfu.edu.sa; 4Chemistry Department, College of Science, King Saud University, Riyadh 11451, Saudi Arabia; anaushad@ksu.edu.sa; 5Department of Organic Chemistry, Krishna University, Machilipatnam 521001, Andhra Pradesh, India; nallakrisharao@gmail.con (N.K.R.); professormandava@gmail.com (M.V.B.R.); 6Department of Materials Science and Engineering, Ulsan National Institute of Science and Technology, UNIST-Gil, Ulsan 44919, Korea; 7Department of Chemical Engineering, A.C Tech Campus, Anna University, Chennai 600025, India; gangireddy.df14@gmail.com; 8Department of Nuclear Physics, Andhra University, Visakhapatnam 530003, Andhra Pradesh, India; t.manoharanaidu@gmail.com

**Keywords:** o-phenylenediamine, nano-ZrO_2_, aryl aldehydes, benzimidazole, XRD

## Abstract

In this article, a zirconia-based nano-catalyst (Nano-ZrO_2_), with intermolecular C-N bond formation for the synthesis of various benzimidazole-fused heterocycles in a concise method is reported. The robustness of this reaction is demonstrated by the synthesis of a series of benzimidazole drugs in a one-pot method. All synthesized materials were characterized using **^1^HNMR, ^13^CNMR, and LC-MS** spectroscopy as well as microanalysis data. Furthermore, the synthesis of nano-ZrO_2_ was processed using a standard hydrothermal technique in pure form. The crystal structure of nano-ZrO_2_ and phase purity were studied, and the crystallite size was calculated from XRD analysis using the Debye–Scherrer equation. Furthermore, the antimicrobial activity of the synthesized benzimidazole drugs was evaluated in terms of Gram-positive, Gram-negative, and antifungal activity, and the results were satisfactory.

## 1. Introduction

Heterocyclic compounds, particularly nitrogen-containing compounds, are extremely important because their presence in many synthetic organic compounds promotes biological activities [1]. The building of the C-N bonds of heterocyclic compounds is principally significant and has been demonstrated to be challenging to medicinal chemists [2]. The introduction of the amine functional group in a precise manner has been broadly examined by chemists and was first reported by Ullmann and Goldberg [3,4]. Given the predominance and relevance of heteroatoms in molecules of interest, C-N bond formation via direct formation of C-N bonds has attracted significant attention. In this context, numerous transition metal complexes, especially of Rh [5], Ir [6], Co [7], Ru [8], and Pd [9], have exhibited outstanding catalytic efficiency towards C-N bond formation reactions [10].

In recent decades, the application of engineered nanoparticles (NPs) has extended to various fields, such as electronics, biomedical applications, and pharmaceuticals. Zirconia (ZrO_2_) NPs are one of the major nanomaterials used for manufacturing refractories, foundry sands, and ceramics. Owing to the desirable mechanical strength, this material is also used in the biomedical field, including biosensors, cancer therapy, implants, joint endoprostheses, and dentistry [11]. However, the wide application of particles has raised concerns about their potential risks to health and the atmosphere, of which ensuring occupational and consumer safety is an essential concern. Thus far, toxicological studies on ZrO_2_ NPs are limited, and the results were controversial. Zirconia (ZrO2) is widely used as a ceramic material and has significant applications in catalysts as well [11,12]. Due to its unique chemical properties (i.e., surface acidity and basicity), it is a valuable catalyst for elimination, dehydration, hydrogenation, and oxidation in chemical transformations. Azole heterocyclic aromatic compounds are divided into two categories, with the first type being older than the second [13]. The conversion of lanosterol to ergosterol due to inhibiting fungal cytochrome P450 3A-dependent C14-α-demethylase depletes ergosterol in the fungal cell membrane [14]. The disturbance of hepatic enzymes may cause the interaction of triazoles with cytochrome P450 [15]. Azole antifungal activity is different for each compound and clinical efficiency may not coincide with in vitro activity [16].

Over the past two decades, the intensity of systemic fungal infections has increased dramatically, mainly because of the *Candida* species. The increasing numbers of immune-compromised patients due to HIV infection, cancer chemotherapy, and organ transplantation are key factors contributing to this occurrence [17,18]. Recently, extended drug research has been conducted because new antifungal agents are critical for treating these life-threatening invasive fungal infections. The main causative fungi targeted by systemic antifungal drugs are *Aspergillus fumigates* and *Candida albicans*, and mycosis drugs for the treatment of patients with deep mycosis should have a wide antifungal spectrum, including at least these microorganisms [19]. 

The benzimidazole nuclei are important synthetic intermediates in drug discovery. Medicinal chemists have been encouraged by the high therapeutic activity of related drugs to synthesize a large number of novel chemotherapeutic agents [20,21,22]. Nitrogen- and sulfur-based heterocyclic compounds show a wide variety of biological activity as antitumor, antibacterial, antifungal, and antiviral agents [23]. The preparation of several biologically active benzimidazole and benzotriazole derivatives has been described in previous reports [24]. Recently, Salauddin et al. reported a review on benzimidazoles, biologically active compounds, which were discussed in terms of antimicrobial, antiviral, anticancer, antiprotozoal, anti-inflammatory, and analgesic activities [25]. Similarly, our group has reported several novel benzimidazole ring systems at the second position of the benzimidazole ring which contain replicas of various biologically active benzimidazoles [26,27,28,29,30,31]. However, it is still necessary to investigate the screening of various novel benzimidazoles. In the current study, some new benzimidazole derivatives have been synthesized as antifungal agents. However, because of the significant biological activity of benzimidazole, it is still difficult to develop a convenient, high-yielding, and environmentally benign procedure for the synthesis of this ring system, which is consistent with our research work on the synthesis of this ring system. [32]. Here, we report a simple synthetic protocol for the preparation of 2-arylbenzimidazoles from o-phenylenediamine (o-PDA) and aromatic aldehydes in the presence of nano-ZrO_2_. Subsequently, we studied antibacterial and antifungal activities using different benzimidazoles, which showed very good results, as shown in Figure 1. 

## 2. Results and Discussion

### 2.1. XRD, FT-IR, and TGA of Nano-ZrO_2_

Figure 2a shows the crystalline nature and phase purity of the resulting nano-ZrO_2_, evaluated by measurements of room temperature XRD. The nano-ZrO_2_ sample diffraction peaks were indexed according to JCPDS card number 80-0965. A tetragonal phase (space group = P42/nmc) was confirmed by the XRD pattern of the investigated sample. In the synthesized powder XRD pattern, no additional peaks were found. Scherrer’s formula was used to calculate the average crystallite size for the most intense diffraction peaks.
(1)$$D=\frac{K\lambda }{\beta \mathrm{Cos}\theta }$$
where *β* (in radians) is the half maximum full width of XRD peaks, *K* = 0.94 is the shape factor, *λ* = 1.54178 Ǻ for Cu-Kα X-rays, *θ* is the diffraction angle (in degrees) corresponding to each plane. Crystallite size (*D*) was found to be 60 ± 5 nm for chemically synthesized nano-ZrO_2_. For (111) planes, the powder diffraction file for cubic and tetragonal ZrO_2_ has the highest peak intensity and the second most intense is (220). By comparing the relative intensities of (111) at 2θ~30° and (220) at 50° in each sample with these data, it appears that the (111) orientation is suppressed and the (110) growth is increased at 600 °C [32,33].

### 2.2. FT-IR

FT-IR of nano-ZrO_2_ showed bending and stretching vibrations of the O-H functional groups due to absorbed water molecules, attributed to the bands noted at 3425 cm^−1^ and 1638 cm^−1^, respectively [29]. The band at 1386 cm^−1^ is ascribed to the absorption of non-bridging O-H groups. The sharp bands at 504 cm^−1^ and 734 cm^−1^ are attributed to the vibration modes of the ZrO3^2−^, and ZrO_2_ groups, which strongly confirm the formation of nano-ZrO_2_, as shown in Figure 2b [34].

### 2.3. Thermal Stability Test (TGA)

Figure 2c shows the TGA analysis that was performed in an inert N_2_ gas atmosphere to determine the thermal degradation of the sample due to the occurrence of air or oxygen atmospheric oxidation in the sample and 92.6% char residue at 800 °C [22]. It is obvious that nano-ZrO_2_ exhibited continuous weight loss, and the weight loss below 300 °C is attributed to the release of physically adsorbed molecules, mainly water as moisture, and the weight loss above 300 °C is due to the desorption of chemically bonded oxygenated groups and the dehydration of surface hydroxyls. Based on the result, nano-ZrO_2_ is highly thermally stable up to 800 °C [23]. Recently, Zhou et al. reported the dispersion behavior of zirconia nanocrystals and their surface functionalization with vinyl group-containing ligands. Our results are similar to their TGA data [35,36,37].

### 2.4. TEM, Size Distribution, SEM, and EDX Analysis of Nano-ZrO_2_

The morphology, size, and shape of the as-synthesized nano-ZrO_2_ were investigated by using HR-TEM. The TEM image of nano-ZrO_2_ showed stretched spherical particles with diameters of ~40–60 nm. Nano-ZrO_2_ has been well known for several years as having a variety of sizes and shapes, however, stretched spherical nano-ZrO_2_, as shown in Figure 3a, is rarely found. The corresponding size distribution histogram of nano-ZrO_2_ is shown in Figure 3b, and the calculated average particle size was about 60 nm, using ImageJ software. Further, the surface morphology of nano-ZrO_2_ was confirmed using SEM analysis, as shown in Figure 3c. The SEM image clearly showed that nano-ZrO_2_ appeared as aggregated spherical surface morphologies. The agglomeration of nano-ZrO_2_ is visible in the SEM images as well. Next, the elemental composition of nano-ZrO_2_ was analyzed by EDX [30]. The host material of nano-ZrO_2_ exhibited three elemental peaks which correspond to zirconium and oxygen at 0.1, 1.98, and 0.56 keV, respectively. From the EDS data, the weight ratio of Zr:O was approximately 77:23 and the spectra suggested that the nano-ZrO_2_ consisted of only Zr and O elements [38].

### 2.5. Synthesis of 4-Methoxy Benzimidazole *(**3a**)*

2-(4-methoxy phenyl) benzimidazole (**3a**) was synthesized by a mixture of o-PDA (1) and 4-methoxy benzaldehyde (**2a**) in the presence of nano-ZrO_2_ at 60 °C. The reaction was performed with 2 mol% of nano-ZrO_2_ in diverse aprotic to protic solvent systems which included dichloromethane (DCM), acetonitrile (ACN), methanol (MeOH), toluene, 1, 4-dioxane, THF, DMF, DMSO, and dry ethanol (entries 1–9, Table 1). Among all the solvents, the dry ethanol was suitable for condensation of 1 and 2a using nano-ZrO_2_ at 60 °C and produced 2-(4-methoxy phenyl) benzimidazoles (**3a**) with 93% yield in 3 h (entry 1, Table 2). A thorough examination of the reaction using the aforementioned solvents revealed lower yields, longer reaction times, and recovered starting material as a result of an incomplete condensation process. Mainly, DCM (45%), CH_3_CN (42%), MeOH (53%), toluene (55%), 1, 4-dioxane (35%), THF (32%), DMF (46%), DMSO (48%) showed poor condensation between o-PDA (1) and 4-methoxy benzaldehyde (**2a**) due to poor product formation. To overcome these practical issues, we chose dry ethanol as an alternative solvent to complete the reaction with good to excellent yields. Less than 12% unreacted starting materials were found in ethanol compared with the other solvents. Hence, the optimization was carried out in ethanol by varying the amount of nano-ZrO_2_. The formation of **3a** was observed to have a high yield and the reaction with 2 mol% of nano-ZrO_2_ and 2.5 mol% of catalyst showed a similar product yield of 92%. Therefore, the reaction was stabilized with 2 mol% of nano-ZrO_2_ in dry ethanol [39,40,41].

Next, we investigated the preparation of **3a** from **1** with **2a**, performed using various commercially available metal nano-catalysts, such as 5% Pd/c, 5% Pd/Al_2_O_3_, 5% Pd/BaCO_3_, 5% CuO nanoparticles (CuO NPs), and 5% Zn(NO_3_)_2_·XH_2_O. Additionally, we conducted a blank experiment without nano-ZrO_2_ in dry ethanol and observed up to 15% monoadduct was formed. In addition, the conversion of **3a** was very poor even at higher temperatures for 8 h, as shown in Table 2.

To check the reusability of nano-ZrO_2_, when the reaction of o-PDA with various substituted aromatic aldehydes was over, the product formed was extracted with ethyl acetate and the catalyst was purified. It was washed with ethyl acetate repeatedly, dried, and reused for the reaction of o-PDA with various aryl aldehydes. The nano-ZrO_2_ was found to be reusable for at least four cycles without any inactivity, and from the fifth cycle the catalytic activity was decreased (Table 3) [41].

The main advantage of nano-ZrO_2_ is that besides its selectivity, it showed moderate recyclability, and it can be recovered through simple filtration, or a decantation method (Table 3). To the best of our knowledge, nano-ZrO_2_ is a recyclable catalyst for the synthesis of benzimidazoles by condensation of **1** and **2a** to produce **3a** with a good yield [10,33]. When the catalyst was recovered and reused without any treatment, the reaction rate gradually decreased (entry 1–5) due to moisture contamination. Therefore, for each successive use, we gently dried the catalyst and reused it. Nano-ZrO_2_ showed moderate catalytic activity after the fifth cycle, and produced **3a** with an 82% yield.

With optimized reaction conditions, the scope of the different aryl aldehydes was examined for the synthesis of various aryl benzimidazoles. Initially, o-PDA was reacted with benzaldehyde, in the presence of 2 mol % of nano-ZrO_2_ and produced 2-phenylbenzimidazole, **3a**, with a 95% yield. Next, the reaction of o-PDA was carried out with various aryl benzaldehydes with electron-donating groups using 2 mol % of nano-ZrO_2_ to produce corresponding aryl benzimidazole in good to excellent yields, **3b**–**3d** (i.e., 93% (**3b**), 91% (**3c**), and 90% (**3d**)), respectively. On the other hand, the reaction of o-PDA with aryl benzaldehydes with electron-withdrawing groups at meta and ortho positions (e.g., 3-Cl and 3-Br) in the presence of 2 mol % of nano-ZrO_2_ produced corresponding aryl benzimidazoles with 84% (**3e**) and 86% (**3f**) yields, respectively, as shown in Table 4.

### 2.6. Antibacterial Screening

Figure 4a shows that the cup–plate screening method is simple for measuring inhibition of microorganisms. Here, we have used this method for the antibacterial screening of the tested compounds. Nutrient agar 2%, peptone 1%, beef extract 1%, sodium chloride 0.5%, and distilled water up to 100 mL were used in culture media. All the constituents were weighed and added to the base medium water. The solution mixture was heated in a water bath for about 1.5 h until a clear solution was obtained and the nutrient medium was sterilized in an autoclave. Antibacterial activity was assessed against Gram-positive bacteria (*Streptococcus pneumonia* (RCMB 010010), *Bacillus subtilis* (RCMB 010067)) and Gram-negative bacteria (*Pseudomonas aeruginosa* (RCMB 010046), *Escherichia coli* (RCMB 010052)). Ampicillin was used as a standard in the assessment of antibacterial activity against Gram-positive bacteria and gentamicin was used as a standard in the assessment of the activity of the compounds tested against Gram-negative bacteria. The results were expressed as the mean inhibition zone in mm ± standard deviation beyond the well diameter (6 mm) produced by microorganisms using 10 mg/mL of samples tested, as shown in Table 5.

We investigated the microbial activity of the titled moiety, and the compounds **3b**, **3c**, **3i**, and **3j** demonstrated the highest active potency against *E. coli*. The compounds **3c**, **3d**, **3g**, and **3i** showed maximum active potency against *P. aeruginosa*. The compounds **3d** and **3f** showed moderate active potency against *S. pneumonia*. The compounds **3e** and **3f** showed good active potency against *B. subtilis*. The rest of the compounds exhibited poor to moderate active potency, as shown in Table 5.

We reported that tested compounds **3b**, **3c**, and **3i** are more potent than other compounds in terms of antifungal activity. The results also showed that alkyl benzimidazoles are more active than alkyl benzotriazoles. All tested compounds in this study, except **3i**, **3b**, and **3c**, exhibited desirable activity on *Candida albicans*. Some *Aspergillus niger*, *Aspergillus flavus*, and *Candida albicans*, which are resistant to fluconazole, were affected by the synthesized compounds **3i**, **3b**, and **3c**, as shown in Table 6.

## 3. Methods and Materials

In this study, zirconium (IV) oxynitrate (ZrO(NO_3_)_2_·xH_2_O hydrate, sodium hydroxide (NaOH) 5% Pd/c, 5% Pd/Al_2_O_3_, 5% Pd/BaCO_3_, and 5% CuO nanoparticles (CuO NPs) were acquired from Sigma-Aldrich (St. Louis, MO, USA). ^1^HNMR and ^13^CNMR spectra were recorded on a Bruker Avance (400 and 100 MHz) spectrometer (Billerica, MA, USA) with tetramethylsilane (TMS) as an internal standard. Thin-layer chromatography using silica gel F254 aluminum sheets was used to monitor the reaction progress. The crystal structure and phase purity of nano-ZrO_2_ were investigated and the crystallite size was calculated using the XRD spectrum Debye–Scherrer equation. SEM (TESCAN, CZ/MIRA I LMH, Brno, Czech Republic) was used to investigate the surface morphology of nano-ZrO_2_. The particle size was measured using TEM (FEI, TECNAI G2 TF20-ST, Brno, Czech Republic). The FT-IR spectrum was recorded using a JASCO, FT/IR-6300 spectrometer (Easton, MD, USA). The basic composition of nano-ZrO_2_ was determined by energy dispersive X-ray (EDX) analysis. Thermogravimetric (TGA) analysis was performed on a Perkin-Elmer (Waltham, MA, USA) instrument at a linear heating rate of 10 °C min^−1^ in a nitrogen environment. The temperature range was maintained at 30 to 800 °C. The GC-MS system used consisted of an Agilent instrument (Carpinteria, CA, USA) Network GC system 5975 C series MS Detector, 7683 B series, and an auto injector and was interfaced with Chemstation Software with an Rxi-624Sil MS (30 m length × 0.32 mm I.D. × 1.8 μm film thickness). Column oven temperature was maintained with a program, i.e., initial temperature 40 °C held for 4 min, ramp 40 °C/min to 200 °C, held for 5 min. The injector temperature was 200 °C, interface temperature was 220 °C, column flow (nitrogen) was 2.0 mL/min, and mass range was 50–300. Acquisition mode was electron ionization (EI) mode and the injected sample volume was 1 μL with split mode (1:25).

### 3.1. Synthesis of Nano-ZrO_2_

The synthesis of nano-ZrO_2_ was processed using a standard hydrothermal technique. Briefly, 0.1 M ZrO(NO_3_)_2_·xH_2_O and 0.2 M NaOH were prepared in DI water. Fifty milliliters of each solution ((ZrO(NO_3_)_2_·xH_2_O:NaOH; 1:1) were mixed and poured into a hydrothermal flask, and heated at 150 °C. After 3 h, the separated nano-ZrO_2_ was centrifuged at 5000 rpm for 30 min using a cooling centrifuge machine, and washed 5 times with dry ethanol. The obtained precipitate was dried at 100 °C for 2 h, calcined at 600 °C, and crushed to obtain a fine powder of nano-ZrO_2_.

### 3.2. General Procedure for the Preparation of 2-Phenyl-1H-benzimidazole *(**3a**–**h**)*

Nano-ZrO_2_ (2 mol %) was added to the solution of o-PDA (1 mmol), and corresponding aryl aldehyde (1 mmol) in dry ethanol (20–25 mL), and stirred at 60 °C. The progress of the reaction was checked by TLC. Upon completion of the reaction, DI water (30–40 mL) was added, and the separated solid mixture was purified using a gravity column to obtain pure products.


***2-Phenyl-1H-benzimidazole* (3a)**


IR (KBr): ν= 3356 (NH), 3051 (CH aromatic), 1459 (C=C) cm^−1^. ^1^HNMR (400 MHz, CDCl_3_): δ ppm = 7.72–7.24 (m 7H, Ar-H), 8.09–7.89 (m, 2H, Ar-H), 11.56 (s, 1H, NH); ^13^CNMR (100 MHz, CDCl_3_) δ ppm: 151.61, 142.06, 135.18, 131.29, 129.23, 128.24, 126.17, 123.67, 122.55, 118.88, 112.58. Molecular formula: C_13_H_10_N_2_.


***2-(4-methoxyphenyl)-1H-benzimidazole* (3b)**


^1^HNMR(400 MHz, CDCl_3_): δ3.70 (d, 3H, OCH_3_),6.12 (s, 1H, NH), 6.94 (d, 2H, aromatic), 6.98 (d, 2H, Ar-H, 7.20 (d, 2H, Ar-H), 7.58 (d, 2H, Ar-H); ^13^CNMR (100 MHz, CDCl_3_) δppm: 162.54, 150.08, 129.81, 128.19, 123.46, 122.34, 114.15, 55.56. LCMS (*m*/*z*): 225 (M^+^ + H). Molecular formula: C_14_H_12_N_2_O.


***2-(3,5-Dimethoxyphenylhenyl)-1H-benzimidazole* (3c)**


IR (KBr): ν = 2884 (C-H, aliphatic), 3056 (C-H, Ar-H), 1521 (C=C), 1619 (C=N) cm^−1^. ^1^HNMR (400 MHz, CDCl_3_): δ ppm = 3.74 (s, 3H, OCH_3_), 3.82 (s, 3H, OCH_3_), 7.18–7.87 (m, 7H, C-H, Ar-H), 12.15 (s, 1H, NH). Molecular formula: C_15_H_14_N_2_O_2_.


***2-(4-Methylphenyl)-1H-benzimidazole* (3d)**


IR (KBr): ν = 3323 (NH), 3059 (C-H, Ar-H), 1448 (C=C), 1622 (C=N) cm^−1^. ^1^HNMR (400 MHz, DMSO): δ (ppm) = 12.83 (s, 1H, NH), 8.08–7.17 (m, 8H, Ar-H), 2.38 (s, 3H, CH_3_), ^13^CNMR (100 MHz, CDCl_3_) δ ppm: 152.94, 141.49, 130.56, 129.04, 128.25, 127.11, 125.83, 123.56, 122.48, and 118.75, 111.90. Molecular formula: C_14_H_12_N_2_.


***2-(3-Chlorophenyl)-1H-benzimidazole* (3e)**


IR (KBr): ν = 3364 (NH), 3053 (CH, Ar-H), 1448 (C=C), 745 (C-Cl). ^1^HNMR (400 MHz, CDCl_3_): δ ppm = 12.54 (s, 1H, NH), 8.10–7.36 (m, 8H, Ar-H); ^13^CNMR (100 MHz, CDCl_3_) δppm: 150.96, 141.80, 136.45, 129.21, 128.76, 127.43, 122.84, 119.08, 111.98. LCMS (*m*/*z*): 194.24. Molecular formula: C_13_H_9_ClN_2_.


***2-(3-Bromophenyl)-1H-benzimidazole* (3f)**


IR (KBr): ν = 3346 (NH), 3050 (C-H, Ar-H) 1434 (C=C) cm^−1^. ^1^HNMR (400 MHz, CDCl_3_): δ ppm: 12.87 (s, 1H, NH), 8.02–7.27 (m, 8H, C-H, Ar-H); ^13^CNMR (100 MHz, CDCl_3_) δ ppm: 150.59, 141.48, 131.63, 129.57, 128.98, 128.15, 127.04. LCMS (*m*/*z*): 273.18 (M^+^ + 2); Molecular formula: C_13_H_9_BrN_2_.


***2-(2-Hydroxy-5-bromophenyl)-1H-benzimidazole* (3g)**


IR (KBr): ν = 3358 (NH), 3055 (C-H, Ar-H), 1442 (C=C) cm^−1^. ^1^HNMR (400 MHz, CDCl_3_): δ ppm = 11.94 (s, 1H, NH), 6.97–7.63 (m, 7H, Ar-H), 8.56 (s, 1H, -OH); ^13^CNMR (100 MHz, CDCl_3_) δ ppm: 154.12, 148.38, 140.06, 132.79, 130.26, 126.44, 123.32, 119.76, 116.08, 115.57. LCMS (*m*/*z*): 289.26 (M^+^ + 2): Molecular formula: C_13_H_9_N_2_BrO.


***2-(3-Nitrophenyl)-1H-benzimidazole* (3h)**


IR (KBr): ν = 3358 (NH), 3086 (CH aromatic), 1516 (N=O), 1431 (C=C) cm^−1^, ^1^HNMR (400 MHz, CDCl_3_): δ (ppm) = 12.78 (s, 1H, NH), 8.14–7.28 (m, 8H, aromatic); ^13^NMR (100 MHz, CDCl_3_) δ ppm: 153.08, 147.53, 136.74, 130.65, 128.42, 126.20, 122.87, 119.94, 119.07, 117.56, 112.08. LCMS (*m*/*z*): 240.12 (M^+^ + H). Molecular formula: C_13_H_9_N_3_O_2_.


***2-(2-bromo-3,4-dimethoxyphenyl)-1H-benzimidazole* (3i)**


IR (KBr): ν = 3358 (NH), 2923 (C-H, aliphatic), 3049 (C-H aromatic), 1440 (C=C) cm^−1^, ^1^HNMR (400 MHz, CDCl_3_): δ ppm = 12.72 (s, 1H, NH), 7.55–7.12 (m, 6H, Ar-H), 3.72 (s, 6H, (OCH_3_)_2_); ^13^CNMR (100 MHz, CDCl_3_) δ ppm: 152.98, 149.21, 147.73, 140.08, 137.47, 124.55, 122.73, 115.54, 111.64, 107.95, 61.09, 54.84. LCMS (*m*/*z*): 334.53 (M^+^ + H). Molecular formula: C_15_H_13_BrN_2_O_2_.


***2-(2-iodo-3, 5-dimethoxyphenyl)-1H-benzimidazole* (3j)**


IR (KBr): ν = 3372 (NH), 2879 (C-H, aliphatic), 3063 (CH aromatic), 1429 (C=C) cm^−1^, ^1^HNMR (400 MHz, CDCl_3_): δ ppm = 12.76 (s, 1H, NH), 7.46–7.24 (m, 4H, Ar-H), 7.08 (s, 1H, Ar-H), 6.94 (s, 1H, Ar-H), 3.65 (s, 6H, (OCH_3_)_2_); ^13^CNMR (100 MHz, CDCl_3_) δ ppm: 158.21, 156.64, 149.55, 139.36, 136.47, 128.46, 126.54, 125.73, 122.75, 115.63, 55.09, 53.64. LCMS (*m*/*z*): 380.22 (M^+^). Molecular formula: C_15_H_13_IN_2_O_2_. The NMR and Mass spectra of Benzimidazoles are available in Appendix A.

### 3.3. Antimicrobial and Antifungal Assays

Compounds were used to determine the antimicrobial activity using the agar well diffusion test method. Agar nutrients (Oxoid Laboratories, UK) for bacteria were subcultured, and Saboroud dextrose agar (Oxoid Laboratories, UK) for fungi was tested. As a positive control for bacterial strains, ampicillin and gentamycin were used. As a positive control for fungi, amphotericin was used. Triplicate plates were made. Bacterial cultures were incubated at 37 °C for 24 h, while fungal cultures were incubated at 25–30 °C for 3–7 days. The inhibition zone determined the antimicrobial activity. Antibacterial activity (Gram-positive: *Streptococcus pneumoniae* (RCMB 010010), *Bacillus subtilis* (RCMB 010067) and Gram-negative: *Pseudomonas aeruginosa* (RCMB 0100 46), *Escherichia coli* (RCMB 010052)) was assessed.

While ampicillin was the standard used to evaluate antibacterial activity against Gram-positive bacteria, gentamicin was the standard used to evaluate the activity against Gram-negative bacteria of the tested compounds. Results were expressed as the mean inhibition zone in mm ± standard deviation beyond the well diameter (6 mm) produced by microorganisms using 10 mg/mL of the samples tested, as shown in Table 1.

### 3.4. Determination of MIC

In triplicate, the sample minimum inhibitory concentration (MIC) was estimated for each of the tested organisms. The nutrient broth was added at varying sample concentrations (1000–0.007 μg/mL) and a loop of the test organism previously diluted to the 0.5 McFarland turbidity standard was introduced into the tubes. To serve as a control, a tube containing broth media was only seeded with the test organisms. Tubes containing cultures of tested organisms were then incubated for 24 h at 37 ° C, while the fungal cultures were incubated at 25–30 °C for 3–7 days. The tubes were then examined for growth using turbidity observations [10].

## 4. Conclusions

In summary, this work demonstrated the design of nano-ZrO2 for the application of nano-catalysis and use for the synthesis of benzimidazole ring systems. The main advantages of the proposed nano-ZrO2 are that it is highly reproducible, easy to handle, recyclable, has less mole %, and no leaching effect is found during the organic conversions. The efficient condensation reaction was carried out between o-PDA (1) and various aryl aldehydes (2) for the preparation of substituted-2-arylbenzimidazoles (3) in the presence of the recyclable nano-ZrO_2_ catalyst. The nano-ZrO_2_ catalyst was prepared from readily available reagents and thoroughly investigated using XRD, TEM, SEM, and FT-IR analyses. 2-Aryl benzimidazoles (3) were obtained in good to excellent yields, mainly from electron-donating functional groups. More importantly, the nano-ZrO_2_ catalyst showed excellent recyclability of up to five cycles in dry ethanol. In addition, dry ethanol works as an eco-friendly solvent to protect the environment without producing any by-products. The titled compounds have exhibited diverse antibacterial activity in terms of growth inhibition of microorganisms. Derivatives of synthesized compounds have shown antibacterial activities, manifested as growth inhibition of different Gram-positive bacteria (*S. aureus*, *B. subtilis*) and two Gram-negative bacteria (*P. aeruginosa*, *E. coli*), while antifungal activity against *Aspergillus niger*, *Aspergillus flavus*, and *Candida albicans* exhibited different active potential. During the diffusion method, the newly synthesized compounds showed good microbiological activity. The results confirmed their strong antibacterial and antifungal activities against various microorganisms. Additionally, we are examining other one-pot multicatalytic reactions based on the adaptable activity of the nano-ZrO_2_ catalyst.

## Figures and Tables

**Figure 1 molecules-26-04219-f001:**
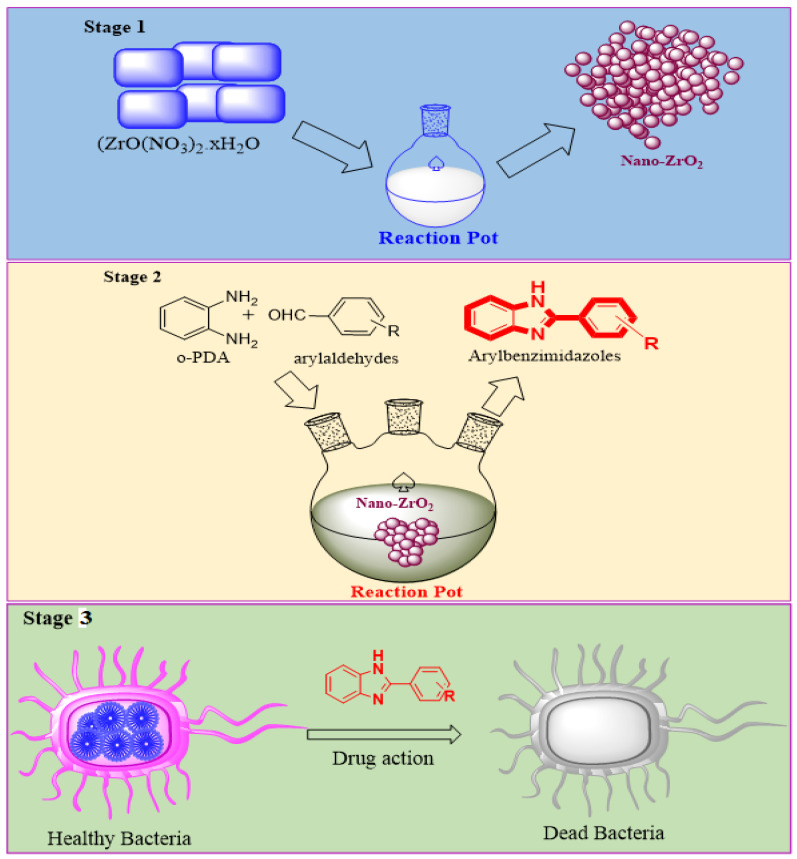
Synthesis of nano-ZrO_2_ catalyst (Stage 1) and aryl benzimidazoles (**3**, Stage 2) and antibacterial and antifungal activities (Stage 3).

**Figure 2 molecules-26-04219-f002:**
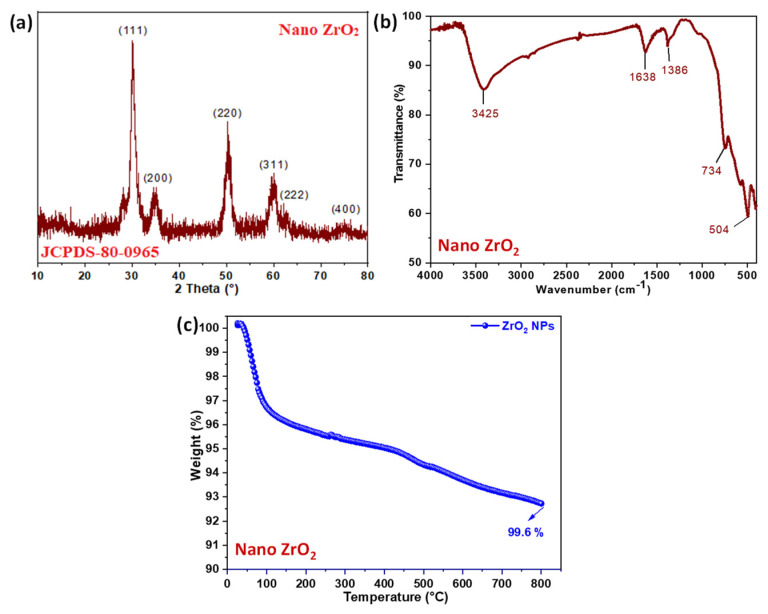
(**a**) XRD, (**b**) FT-IR, and (**c**) TGA of nano-ZrO_2_.

**Figure 3 molecules-26-04219-f003:**
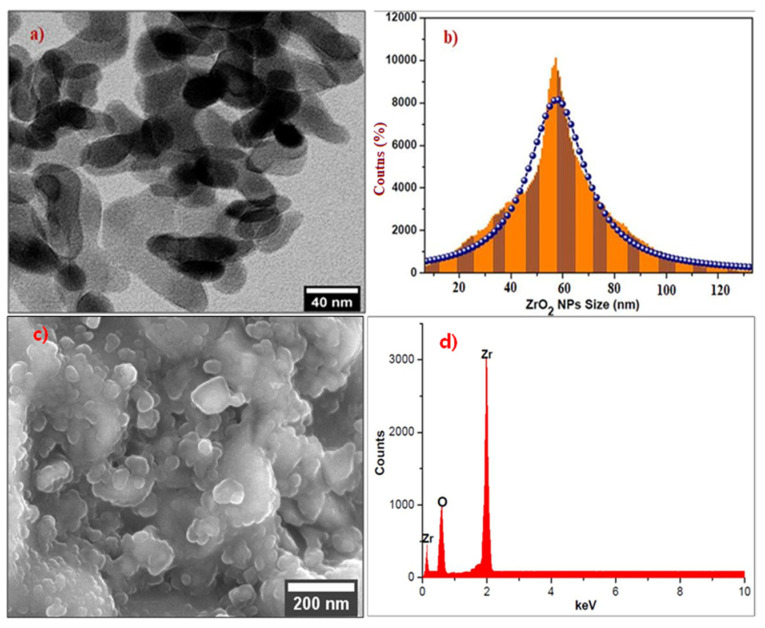
(**a**) TEM, (**b**) distribution histogram, (**c**) SEM, (**d**) EDX of nano-ZrO_2_.

**Figure 4 molecules-26-04219-f004:**
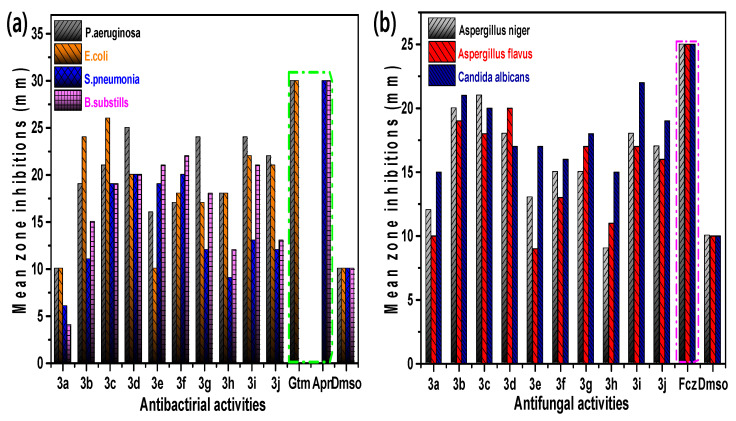
The antibacterial (**a**) and antifungal (**b**) activities of aryl benzimidazole derivatives.

**Table 1 molecules-26-04219-t001:** Optimization study by solvent.

Entry	Nano-ZrO_2_ Mol %	Solvent	Time (h)	Yield %
1	2.0	DCM	3	45
2	2.0	CH_3_CN	3	42
3	2.0	MeOH	3	53
4	2.0	Toluene	3	55
5	2.0	1,4-dioxane	3	35
6	2.0	THF	3	32
7	2.0	DMF	3	46
8	2.0	DMSO	3	48
9	0.5	Dry ethanol	3	63
10	1.0	Dry ethanol	3	79
11	1.5	Dry ethanol	3	86
12	2.0	Dry ethanol	3	92
13	2.5	Dry ethanol	3	92

**Table 2 molecules-26-04219-t002:** Catalyst activity comparison ^[a]^.

S. No	Catalyst Mol %	Temp [°C]	Time [h]	Conv [%] ^[b]^	Product Yield [%] ^[c] [e]^		
					3a	Monoadduct	Starting Materials
1	2% nano-ZrO_2_	60	3	>97	93	03	0
2	5% Pd/C ^[d]^	60	3	65	55	30	15
3	5% Pd/Al_2_O_3_ ^[d]^	60	3	58	50	35	15
4	5% Pd/BaCO_3_ ^[d]^	60	3	52	45	30	20
5	5% CuO NP ^[d]^	60	3	55	55	34	11
6	5% Zn(NO_3_)_2_ XH_2_O ^[d]^	60	3	52	42	43	15
7	No catalyst (blank) ^[d]^	60	3	00	--	15	85

^[a]^ A solution of o-PDA (1mol) and 4-MeO-benzaldehyde (1.1 mol) in dry ethanol (20 mL) was heated at 60 °C under nitrogen for 3h in the presence of various catalysts, ^[b]^ determined by gas chromatography, ^[c]^ isolated yield, ^[d]^ commercially available catalysts, ^[e]^ zirconium was not detected in the filtrate with inductively coupled plasma (ICP) analysis.

**Table 3 molecules-26-04219-t003:** Recycling of nano-ZrO_2_ for synthesis of 4-OMe benzimidazole (**3a**) ^[a]^.

Use	Temp [°C]	Time (h)	Yield % ^[b]^
1	60	3	93
2	60	3	92
3	60	3	90
4	60	3	85
5	60	3	83

^[a]^ A solution of o-PDA (1 mol) and 4-MeO-benzaldehyde (1.1 mol) in dry EtOH (20 mL) was heated at 60 °C under nitrogen for 3 h in the presence of 2 mol% of nano-ZrO_2_, ^[b]^ determined by gas chromatography.

**Table 4 molecules-26-04219-t004:** Nano ZrO_2_ catalysed synthysis of 2-aryl benzimidazoles ^a^.

Entry	o-PDA(1)	Aryl Aldehyde(2)	Product(3)	Yield(%) ^b^	M.P(°C)
**3a**	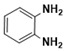	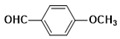	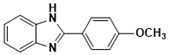	93	226–228(225–226) ^c^
**3b**	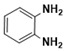	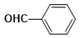	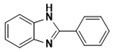	95	287–289(290–292) ^c^
**3c**	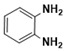	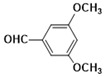	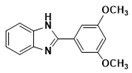	91	246–248
**3d**	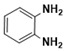	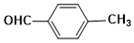	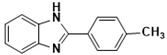	91	262–264(268–270) ^c^
**3e**	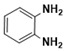	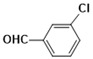	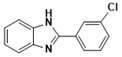	84	233–234(234–236) ^d^
**3f**	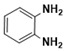	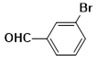	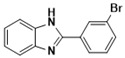	86	245–246
**3g**	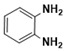	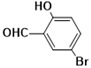	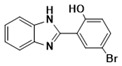	85	252–254(256–257) ^d^
**3h**	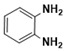	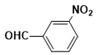	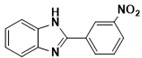	82	201–292(203–204) ^e^
**3i**	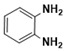	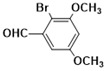	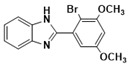	87	254–255
**3j**	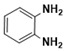	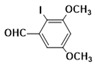	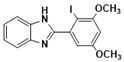	85	278–280

^a^ A solution of o-PDA (1 mol), and aryl benzaldehyde (1.1 mol), in dry. ethanol (20 mL) was heated at 60 °C in nitrogen gas for 3 h in the presence of Nano ZrO_2_ [2 mol%], ^b^ determined by GC, ^c^ [12], ^d^ [13], ^e^ [14]. All the products were showed satisfactory spectroscopic (IR, ^1^HNMR, & ^13^CNMR) analyses data.

**Table 5 molecules-26-04219-t005:** Antibacterial activity of benzimidazole derivatives.

Compound Code	Zone of Inhibition (mm)			
	Bacteria			
	*P. aeruginosa*	*E. coli*	*S. pneumonia*	*B. subtilis*
**3a**	10	10	06	04
**3b**	19	24	11	15
**3c**	21	26	19	19
**3d**	25	20	20	20
**3e**	16	10	19	21
**3f**	17	18	20	22
**3g**	24	17	12	18
**3h**	18	18	09	12
**3i**	24	22	13	21
**3j**	22	21	12	13
gen-gentamicin	30	30	---	---
am-penicillin	---	---	30	30
DMSO	10	10	10	10

**Table 6 molecules-26-04219-t006:** Antifungal activity of benzimidazole derivatives.

Entry	*Aspergillus niger*	*Aspergillus flavus*	*Candida albicans*
**3a**	12	10	15
**3b**	20	19	21
**3c**	21	18	20
**3d**	18	20	17
**3e**	13	09	17
**3f**	15	13	16
**3g**	15	17	18
**3h**	09	11	15
**3i**	18	17	22
**3j**	17	16	19
fluconazole	25	25	25
DMSO	10	10	10

## Data Availability

Not applicable.

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
