# Peer review of "Reusable Nano-Zirconia-Catalyzed Synthesis of Benzimidazoles and Their Antibacterial and Antifungal Activities"

_molecules, 2021, doi:10.3390/molecules26144219_

Round 1

Reviewer 1 Report

This work reports a new nano-ZrO2, which has been applied for synthesis of benzimidazoles and further studied their antibacterial and antifungal activities. The results are interesting but there are still many issues that need to be addressed. 1. The abstract should be rewritten. Currently, it looks more like an experimental report. The main findings and the contributions of this work should be present. 2. In the second paragraph of Introduction, the current synthesis technology of ZrO2 should be introduced to readers. Why the authors try to develop a new synthesis method for ZrO2? 3. The advantages of new nano-ZrO2 should be summarized in the conclusion section. 4. All the original IR, 1HNMR & 13CNMR spectra in the section 2.1 should be provided in the supporting information. Also, several mistakes of 13NMR have appeared in section 2.1. Please correct them. 5. In section 2.4: why the separation of nano-ZrO2 was so restrict? Could it be separated by higher speed and less time? Also, the “dry. Ethanol” is suggested to be revised as “dry ethanol”. Please check other places in the manuscript. The dot after dry would misunderstand reader. 6. The standard peaks of JCPDS card number 80-0965 should be presented in Fig. 2a. 7. “the weight loss below 300 °C is generally assigned to the release of physically adsorbed molecules, mainly water, methanol” Where is the methanol from? The authors did not use the methanol to synthesize the ZrO2. 8. What is the unit of Fig. 3b? How could the authors count so many dots? Perhaps the unit changing to percentage is more suitable. 9. Please provide a control experiment on the synthesis of benzimidazoles to compare the activity of commercial tetragonal ZrO2 and nano-ZrO2 here. Commercial ZrO2 has been widely used in different research areas. Why the authors try to prepare a new nano zirconia here? 10. The language should be revised and many typos (e.g. the subscript of ZrO2) should be corrected.

Author Response

Respected Reviewer,

Greetings,

I cordially thank you for your kind review of our manuscript. I have answered all your valuable questions and incorporated in yellow color the required information in the manuscript wherever it is needed. Also, I have made additional file as supporting information for the some of the important benzimidazole molecules.

Regards,

Ravi Kumar Cheedarala

Reviewer 1

Comments and Suggestions for Authors

This work reports a new nano-ZrO2, which has been applied for synthesis of benzimidazoles and further studied their antibacterial and antifungal activities. The results are interesting but there are still many issues that need to be addressed.

  1. The abstract should be rewritten. Currently, it looks more like an experimental report. The main findings and the contributions of this work should be present.

Ans: we have modified and rewritten the abstract of the manuscript.

Abstract:

In this article, Zirconia based Nano catalyst (Nano-ZrO2), intermolecular C-N bond formation for the synthesis of various benzimidazoles-fused heterocycles in a concise method has been reported. The robustness of this reaction is demonstrated by the synthesis of a series of benzimidazoles drugs in one-pot method. All synthesized materials were characterized using 1HNMR, 13NMR, and LC-MS spectroscopy as well as microanalysis data.  Furthermore, the synthesis of Nano ZrO2 was processed using a standard hydrothermal technique in pure form. The crystal structure of Nano ZrO2, and phase purity was studied, and the crystallite size was calculated from the XRD analysis using the "Debye Scherrer" equation. Besides, the antimicrobial activity of synthesized benzimidazole drugs was evaluated using gram-positive, gram-negative, and anti-fungal and obtained satisfactory results.

  1. In the second paragraph of Introduction, the current synthesis technology of ZrO2 should be introduced to readers. Why the authors try to develop a new synthesis method for ZrO2?

Ans: Since few decades, the application of engineered nanoparticles (NPs) has been extended in various fields, such as electronics, biomedical applications, and pharmaceuticals. Zirconia (ZrO2) NPs are one of the major nanomaterials used for manufacturing refractories, foundry sands, and ceramics. Owing to the desirable mechanical strength, this material is also used in biomedical field, including biosensors, cancer therapy, implants, joint endoprostheses, and dentistry [11]. However, the wide application of particles has raised concern on their potential risks to health and atmosphere, of which ensuring occupational and consumer safety is an essential concern.  So far, toxicological studies on ZrO2 NPs are limited, and the results were controversial.

  1. The advantages of new nano-ZrO2 should be summarized in the conclusion section.

Ans:

  1. All the original IR, 1HNMR & 13CNMR spectra in the section 2.1 should be provided in the supporting information. Also, several mistakes of 13NMR have appeared in section 2.1. Please correct them.

Ans: We have attached the 1H-NMR and 13 NMR spectra were added in supporting information and rectified mistakes in the 13 NMR data. (Please find yellow color highlights)

  1. In section 2.4: why the separation of nano-ZrO2 was so restrict? Could it be separated by higher speed and less time? Also, the “dry. Ethanol” is suggested to be revised as “dry ethanol”. Please check other places in the manuscript. The dot after dry would misunderstand reader.

Ans: The nano-ZrO2 particles were light weight if we used high speed they may be strongly adhered on the surface of filter paper, then recovery of 100% is not possible. To avoid such practical issues we separated slow filtration or centrifuge technique.

We changed the from dry. Ethanol to dry. Ethanol in the whole manuscript.

  1. The standard peaks of JCPDS-80-0965 should be presented in Fig. 2a.

Ans: We have incorporated JCPDS card number 80-0965 on the XRD, Fig.2a.

  1. “the weight loss below 300 °C is generally assigned to the release of physically adsorbed molecules, mainly water, methanol” Where is the methanol from? The authors did not use the methanol to synthesize the ZrO2.

Ans: Its mistake, we removed the methanol word.

  1. What is the unit of Fig. 3b? How could the authors count so many dots? Perhaps the unit changing to percentage is more suitable.

Ans:  we have changed the units in Fig.3b as Counts (%).

  1. Please provide a control experiment on the synthesis of benzimidazoles to compare the activity of commercial tetragonal ZrO2 and nano-ZrO2 here. Commercial ZrO2 has been widely used in different research areas. Why the authors try to prepare a new nano zirconia here?

Ans: We have performed the control experiments using commercial ZrO2 and we incorporated into the manuscript. Please find the entries 6 & 7 of Table 2.

  1. The language should be revised and many typos (e.g. the subscript of ZrO2) should be corrected.

Ans: we have revised and checked the typos in the whole manuscript.

Reviewer 2 Report

This manuscript describes the preparation and characterization of the zirconium-based catalyst for benzimidazoles synthesis though the coupling of o-phenylenediamine with aromatic aldehydes. The antibacterial properties of the final products were investigated. The manuscript is, in general, well written and clear. The target products (benzimidazole derivatives) are of interest as building blocks in medicinal chemistry, thus the motivation for this study is fully justified. I recommend acceptance of the manuscript after correction of the following issues:

1) Please reorder the experimental part in the way that follows the study, i.e., first the synthesis of the catalyst (currently section 2.4), then the catalytic protocol and characterization of the products (currently 2.1), then bioactivity studies (currently 2.2 and 2.3).

2) There is no description of the GC chromatographic analysis (device, type of the column, conditions etc). This is important since the products look to be rather heavy as for GC method. It would be nice if the Authors add example chromatograms in the Supporting Info with the peaks assigned to the respective substrate(s) and products.

3) There is only one citation of the recent work on the benzimidazole synthesis (phrase “The preparation of several bio-logically active benzimidazole, and benzotriazole derivatives has been described in pre-vious reports.[24]” in the Introduction). The Authors should expand this section by adding more citations to the benzimidazole synthesis and discuss them briefly. The brief search over the literature reveals plenty of very recent studies, including those using heterogeneous recyclable catalysts. The same concerns the catalytic section: please consider adding some comparative table which summarize the main parameters (yields etc) of the present method and literature ones.

4) It looks like no blank test (the test in the absence of the catalyst) is reported. Please check and add it, explaining the products obtained (if any). The blank test is the formal procedure which aim is to understand and exclude possible background catalyst-free activity.

5) As there is always a possibility of leaching of metal into the solution, the tests using various simple ethanol-soluble zirconium catalysts must be performed. What happens if one uses the starting material, ZrO(NO3)2, or similar compounds, e.g. Zr(NO3)4, as catalysts?

6) What is the conversion in the Table 2? There are two components (o-pda and aldehyde), which one was used to calculate the conversion?

7) The mechanism depicted in the Scheme 2 seems to be too speculative and not supported by experimental evidence. The reaction, most likely, starts from the reaction of aldehyde with amine to produce an imine intermediate. This process is known to proceed quickly and does not require any catalyst. Further, from the proposed mechanism it is not clear what is the necessity of ZrO2 nanoparticle itself (i.e. why it could not be replaced with simple zirconium salt). In turn, the dehydrogenation step should require some catalyst. Please carefully reconsider this mechanism and confirm or refuse it by performing the additional tests (briefly study the dehydrogenation activity of nano-ZrO2 etc).

Author Response

Respected Reviewer,

Greetings,

I cordially thank you for your kind review of our manuscript. I have answered all your valuable questions and incorporated in blue color the required information in the manuscript wherever it is needed. Also, I have made additional file as supporting information for the some of the important benzimidazole molecules.

Reviewer 2

Comments and Suggestions for Authors

This manuscript describes the preparation and characterization of the zirconium-based catalyst for benzimidazoles synthesis though the coupling of o-phenylenediamine with aromatic aldehydes. The antibacterial properties of the final products were investigated. The manuscript is, in general, well written and clear. The target products (benzimidazole derivatives) are of interest as building blocks in medicinal chemistry, thus the motivation for this study is fully justified. I recommend acceptance of the manuscript after correction of the following issues:

1) Please reorder the experimental part in the way that follows the study, i.e., first the synthesis of the catalyst (currently section 2.4), then the catalytic protocol and characterization of the products (currently 2.1), then bioactivity studies (currently 2.2 and 2.3).

Ans: Thank you, we have re-ordered the sections 2.1 to 2.4

2) There is no description of the GC chromatographic analysis (device, type of the column, conditions etc). This is important since the products look to be rather heavy as for GC method. It would be nice if the Authors add example chromatograms in the Supporting Info with the peaks assigned to the respective substrate(s) and products.

Ans: The GC-MS system used, consisted Agilent make, Network GC system 5975 C series MS Detector, 7683 B series, Auto Injector and inter-faced with Chemstation Software with a Rxi-624Sil MS (30 m length x 0.32 mm I.D. x 1.8 μm film thickness). Column oven temperature was maintained with program ie., Initial temperature 40°C held for 4 min, ramp @40°C /min to 200°C held for 5 min. The injector temperature is 200°C, Interface temperature is 220°C, Column flow (Nitrogen) is 2.0 mL/min, and selecting mass range is 50-300. Acquisition mode is SIM and the injected sample volume was 1 μL with split mode (1:25).

3) There is only one citation of the recent work on the benzimidazole synthesis (phrase “The preparation of several bio-logically active benzimidazole, and benzotriazole derivatives has been described in pre-vious reports.[24]” in the Introduction). The Authors should expand this section by adding more citations to the benzimidazole synthesis and discuss them briefly. The brief search over the literature reveals plenty of very recent studies, including those using heterogeneous recyclable catalysts. The same concerns the catalytic section: please consider adding some comparative table which summarize the main parameters (yields etc) of the present method and literature ones.

Ans: we have cited a recent benzimidazole literature examples.

The preparation of several biologically active benzimidazole, and benzotriazole derivatives has been described in previous reports.[24] Recently, Salauddin et. al. have reported a review on benzimidazoles: a biologically active compounds where they have been discussed on anti-microbial, antiviral, anti-cancer, anti-protozoal, anti-inflammatory and analgesic activities. Similarly, our group has been reported several novel benzimidazole ring systems at the second position of the benzimidazole ring which contain replica of various biologically active benzimidazoles.

  1. Salahuddin, Shaharyar, Avijit Mazumder, Arabian Journal of Chemistry, 2017, 10, S157-S173.
  2. K Ramaiah, P. K. Dubey, R. Kumar, R. K. Cheedarala, J. S. Grossert and D L Hooper, Synthetic Comm., 2001, 31, 3439-3446.
  3. P. K. Dubey, R. K. Cheedarala, J Ramanatham and R. Kumar, Indian J. Heterocyclic Chemistry, 2000, 9, 259-262.
  4. P. K. Dubey, R. K. Cheedarala, and B. Balaji, Indian J. Heterocylic Chemistry, 2002,12, 95-98
  5. P. K. Dubey, R. K. Cheedarala, and A. Naidu, Indian J. Chemistry, 2003, 42B, 931- 934
  6. P. K. Dubey, R. K. Cheedarala, A. Naidu, and P. V. V. P. Reddy, Indian J. Chemistry, 2003, 42B, 1701-1705.
  7. K. Ramaiah, P. K. Dubey, J. Ramanatham, R. K. Cheedarala, and J. S. Grossert, Indian J. Chemistry, 2003, 42B, 1765-1767.

4) It looks like no blank test (the test in the absence of the catalyst) is reported. Please check and add it, explaining the products obtained (if any). The blank test is the formal procedure which aim is to understand and exclude possible background catalyst-free activity.

Ans: W have performed a blank tests and reported in Table 2.

5) As there is always a possibility of leaching of metal into the solution, the tests using various simple ethanol-soluble zirconium catalysts must be performed. What happens if one uses the starting material, ZrO(NO3)2, or similar compounds, e.g. Zr(NO3)4, as catalysts?

Ans: Zirconium was not detected in the filtrate with inductively coupled plasma (ICP) analysis.

6) What is the conversion in the Table 2? There are two components (o-pda and aldehyde), which one was used to calculate the conversion?

Ans: The conversion was calculated with respect to o-PDA as starting material.

7) The mechanism depicted in the Scheme 2 seems to be too speculative and not supported by experimental evidence. The reaction, most likely, starts from the reaction of aldehyde with amine to produce an imine intermediate. This process is known to proceed quickly and does not require any catalyst. Further, from the proposed mechanism it is not clear what is the necessity of ZrO2 nanoparticle itself (i.e. why it could not be replaced with simple zirconium salt). In turn, the dehydrogenation step should require some catalyst. Please carefully reconsider this mechanism and confirm or refuse it by performing the additional tests (briefly study the dehydrogenation activity of nano-ZrO2 etc).

Ans: The plausible mechanism is projected for the preparation of benzimidazoles via the condensation of o-PDA, and aryl aldehydes using Nano-ZrO2 as a reusable catalyst where the Nano-ZrO2 is activated from conduction band (CB) to valence band (VB). Mainly, at the initial stage, the Nano ZrO2 was reacted with aryl aldehyde (1) to form an aldehyde-ZrO2 complex (S-12). Next, aldehyde-ZrO2 complex, 2 was condensed with o-PDA (3) by donation of loan pair of electrons from one of the amines to form an intermediate unstable complex (S-2, 4). Complex 4 was converted into imine complex (S-35) by the elimination of H2O molecule. Subsequently, the loan pair of the second amine was attacked on the imine system to form a temporary unstable ionic complex (S-46). Then, the elimination of H2 from the hydrogen-rich ring system (5), and rearranged into a stable benzimidazole ring system (S-5, 7). The aryl benzimidazole was rearranged by proton transfer through the tautomeric form of 7, as shown in Scheme 2. [40, 41]

Round 2

Reviewer 1 Report

The authors have addressed most of the questions. However, the following issues should be further revised:

  1. The origninal data (e.g. representive GC data, Qestion 2 from reviewer 2; IR data, Qestion 4 from reviewer 1 ) were not provided by the authors. These original data should be provided in the supporting information.
  2. The Figure 3 is missing in the revised manuscript. Please add it. 
  3. "Please provide a control experiment on the synthesis of benzimidazoles to compare the activity of commercial tetragonal ZrO2 and nano-ZrO2 here." The authors replied it has been provided in Table 2. However, there was not any results related to this question. Please address this question again. 
  4. Table 2. Why there was 25% conversion for the blank test without any catalyst? Please explain it. 

Author Response

Greetings,

I cordially thank you for your kind review of our manuscript. I have answered all your valuable questions and incorporated in blue color the required information in the manuscript wherever it is needed. Also, I have made additional file as supporting information for the some of the important benzimidazole molecules.

Comments and Suggestions for Authors

The authors have addressed most of the questions. However, the following issues should be further revised:

The origninal data (e.g. representive GC data, Qestion 2 from reviewer 2; IR data, Qestion 4 from reviewer 1 ) were not provided by the authors. These original data should be provided in the supporting information.

Ans: Thank you for you question. We provided original 1H and 13C NMR data for the new compounds. In the case of the GC and FT-IR original data is missing due to instruments software issues. If still wanted to add, then we want to remove from the Table 2 about GC information.

The Figure 3 is missing in the revised manuscript. Please add it.

Ans: we incorporated the Figure 3 in the manuscript.

"Please provide a control experiment on the synthesis of benzimidazoles to compare the activity of commercial tetragonal ZrO2 and nano-ZrO2 here." The authors replied it has been provided in Table 2. However, there was not any results related to this question. Please address this question again.

Ans: We have added the write up about the blank experiment information in the page 9 and highlighted in blue color for your reference.

Table 2. Why there was 25% conversion for the blank test without any catalyst? Please explain it.

Ans: Sorry for the inconvenience. Its typo error. Now, I removed from the table 25%.

Reviewer 2 Report

The authors significantly improved their initial version of the manuscript, adding the relevant data. However, some of their answers require further discussion.

1) The Authors wrote in the GC description section 2: “...selecting mass range is 50-300. Acquisition mode is SIM”. This combination of phrases is senseless because SIM (Single Ion Monitoring) mode of an MS detector presumes monitoring of some certain m/z signals, in contrast to full range monitoring ("50-300" m/z scan, as the Authors wrote). Which mode was actually used? Was MS used for quantification? Again, it highly desirable to add some example chromatograms to the Supplementary Information.

2) The Authors have added more explanations regarding the reaction mechanism. However, the problem remains the same: this mechanism looks not probable and is not supported neither by experimental, nor theoretical evidence. Speculatively, in the case the catalyst was added to the mixture of o-PDA and aldehyde (as it is written in the section 2.2), the reaction mechanism should start from formation of an imine (a Schiff base) from phenylenediamine and aldehyde. This type of reaction is known to proceed quickly and with quantitative yields in the absence of a catalyst. It is totally unclear why the aldehyde should coordinate to ZrO2 first, and also it is not clear how dehydrogenation occurs in the non-catalytic pathway (Scheme 2). The Table 2 shows some “Mono Adduct”, which is never mentioned in the main text. In case it is an imine, its formation is clearly seen in the blank text (Table 2), what confirms that the reaction between amine and aldehyde proceeds in the absence of the catalyst. Please carefully reconsider the mechanism. If the particular steps cannot be confirmed, I suggest elimination of this proposed mechanism (Scheme 2) as well as the respective discussion (section 3.6) at all because it will just confuse readers.

Author Response

Reviewer 2

Greetings,

I cordially thank you for your kind review of our manuscript. I have answered all your valuable questions and eliminated the required information in the manuscript wherever it is needed.

Comments and Suggestions for Authors

The authors significantly improved their initial version of the manuscript, adding the relevant data. However, some of their answers require further discussion.

1) The Authors wrote in the GC description section 2: “...selecting mass range is 50-300. Acquisition mode is SIM”. This combination of phrases is senseless because SIM (Single Ion Monitoring) mode of an MS detector presumes monitoring of some certain m/z signals, in contrast to full range monitoring ("50-300" m/z scan, as the Authors wrote). Which mode was actually used? Was MS used for quantification? Again, it highly desirable to add some example chromatograms to the Supplementary Information.

Ans: The instrument acquisition mode is electron impact ionization source (EI) mode, we regret to inform you that we mistakenly mentioned.

2) The Authors have added more explanations regarding the reaction mechanism. However, the problem remains the same: this mechanism looks not probable and is not supported neither by experimental, nor theoretical evidence. Speculatively, in the case the catalyst was added to the mixture of o-PDA and aldehyde (as it is written in the section 2.2), the reaction mechanism should start from formation of an imine (a Schiff base) from phenylenediamine and aldehyde. This type of reaction is known to proceed quickly and with quantitative yields in the absence of a catalyst. It is totally unclear why the aldehyde should coordinate to ZrO2 first, and also it is not clear how dehydrogenation occurs in the non-catalytic pathway (Scheme 2). The Table 2 shows some “Mono Adduct”, which is never mentioned in the main text. In case it is an imine, its formation is clearly seen in the blank text (Table 2), what confirms that the reaction between amine and aldehyde proceeds in the absence of the catalyst. Please carefully reconsider the mechanism. If the particular steps cannot be confirmed, I suggest elimination of this proposed mechanism (Scheme 2) as well as the respective discussion (section 3.6) at all because it will just confuse readers.

Ans: To stop confusion the readers, we want to remove the mechanism part. Thank you.